# Occupational health risk of farmers: A qualitative study with the agriculture society of trinidad and tobago and the ministry of agriculture, land and fisheries

Barry Parasram[1,2], Avishek Choudhury [1]*

1 Industrial and Management Systems Engineering, West Virginia University, Morgantown, West Virginia 26506, United States of America, 2 Cipriani College of Labour and Cooperative Studies, Valsayn, Trinidad & Tobago

* avishek.choudhury@mail.wvu.edu

## Abstract

The agriculture sector is significantly affected by pesticide toxicity, leading to fatal and chronic diseases. The objective of this research is to explore the perception of the farmers, the representatives of the Agriculture Society of Trinidad and Tobago (ASTT), and the officials of the Ministry of Agriculture, Land and Fisheries (MALF) on the protection of farmers, particularly on the exposure to pesticides in Trinidad and Tobago. We conducted face-to-face, semi-structured interviews with 32 individuals from the MALF and ASTT. Interviews were analyzed using inductive thematic analysis and deductive reasoning. Responses reveal six themes: (a) Barriers to Proper Use of Personal Protective Equipment (PPE); (b) Challenges in Safe Chemical Use and Disposal Practices; (c) Health Impacts of Unsafe Chemical Exposure, (d) Challenges in Farmer Training and Adoption of Safety Practices; (e) Lack of Policies and Monitoring for Farmer Safety and Chemical Use; (f) Absence of Quality Control and Unsafe Practices in Local Produce. A consistent finding suggested an interplay between limited safety awareness, entrenched cultural practices, and the absence of rigorous policy enforcement. Farmers, Ministry of Agriculture officials, and industry representatives acknowledge that while isolated instances of good practice exist, these are overshadowed by regulatory oversight. Farmers remain at significant risk of adverse health outcomes, and consumers may face potential hazards from unmonitored chemical residues in local food. Our research highlights the urgent need for better safety regulations, more effective training, and the establishment of monitoring systems to protect the health of farmers and ensure safe agricultural practices. By shedding light on these issues, this study paves the way for developing targeted interventions to improve farmer health and safety standards in Trinidad and Tobago's agricultural sector.

**Data availability statement:** The datasets generated during and/or analyzed during the current study is available in Dyrad data repository at https://doi.org/10.5061/dryad.x0k6djhwg.

**Funding:** The author(s) received no specific funding for this work.

**Competing interests:** The authors have declared that no competing interests exist.

## Introduction

Pesticides are environmental contaminants that are specifically designed to control pests and diseases. However, these chemicals, which are intended to uplift agricultural practices, have been responsible for an estimated 200,000 acute poisoning deaths every year, mainly affecting farmers in low and middle-income countries (LMICs) [1]. The World Health Organization (WHO) estimates that there is a further one million unintentional pesticide poisoning incidents occurring every year, leading to 20,000 deaths [2]. The agriculture industry, which accounts for over a billion workers, is significantly affected by pesticide toxicity [3,4]. There are significant concerns over the toxic effects of pesticides on humans where linkages to neurological disease [5] have been noted, making farmers susceptible to ailments including lymphocytic leukemia, non-Hodgkin's lymphoma, lung cancer, skin cancer, [6,7] and other chronic diseases [7]. Studies have also outlined exposure to pesticides as an increased risk of Parkinson's disease [8] and cognitive impairment [9]. For instance, reductions in cholinesterase activity—an indicator of neurotoxic effects—have been observed in pesticide applicators [10]. Chronic exposure has been associated with dermatitis, respiratory issues, and neurological deficits, as reported among Thai rice farmers [11]. The repercussions of pesticide overuse extend beyond individual farmers to entire communities and ecosystems. Contamination of water sources and threats to food safety have been documented in Bangladesh [12] and other regions with high pesticide reliance [13]. Such community-wide health risks accentuate the need for policies safeguarding environmental integrity and farmer well-being.

Safety training is consistently cited as a critical intervention for reducing pesticide-related risks. Farmers who receive targeted training are more likely to engage in safer pesticide handling and application [14]. Ren and Jiang reported that agricultural cooperatives' training in Shandong Province, China, decreased pesticide overuse among farmers, emphasizing the role of structured training in promoting safe practices [15]. Integrated Pest Management (IPM) training has emerged as an effective strategy for reducing pesticide exposure. IPM training in Uganda resulted in lower organophosphate exposure among farmers, as measured by acetylcholinesterase (AChE) blood levels [16]. Furthermore, Son et al. found that after two years of IPM training in Burkina Faso, farmers used less pesticide and experienced reduced exposure, indicating the long-term benefits of such educational interventions [17].

However, access to practical training remains limited in many LMICs. Nwadike et al. highlighted that many farmers in Northern Nigeria lack sufficient occupational safety training, contributing to unsafe pesticide application practices [18]. LMICs often lack safety training due to limited resources [19]. This finding is echoed by Çevik et al., who reported that farmers' knowledge, attitudes, and practices regarding pesticide safety are often insufficient, indicating a pressing need for targeted training interventions [20]. Despite some awareness of the risks associated with pesticide use, many farmers fail to implement safe practices, suggesting that knowledge alone is not enough to change behavior [21]. In Tanzania, farmers were aware of how pesticides could be absorbed, yet they did not consistently translate this knowledge

into safer practices [22]. Proper use of PPE is also essential to minimize pesticide exposure. Yet research indicates that farmers often do not use PPE correctly, do not have reliable access to it, or discontinue its use [23–25]. Even when farmers are conscious of the risks, their PPE usage may not effectively reduce exposure if the equipment is of low quality or if training on its proper use is inadequate [23].

Governmental policies and regulations are essential for farmers' access to safe pesticides and training resources. In LMICs, policy flaws also allow the inappropriate use of toxic pesticides banned in the European Union (EU) and the United States of America (US) [26]. Phung et al. emphasized the need for improved pesticide regulations in Vietnam to enhance farmworker safety, highlighting that many farmers are unfamiliar with safety practices and regulations [27]. Similarly, Fuhrimann et al. pointed out that the lack of adequate legislative frameworks in LMICs contributes to the high prevalence of pesticide-related health issues [28]. Therefore, strengthening regulatory frameworks and providing compliance support is vital for improving pesticide safety among farmers.

This study focuses on farmers' health and occupational hazards in Trinidad and Tobago. The exposure to pesticides in this nation is significant, and the farmers experience acute occupational pesticide poisoning resulting from little to no training on safe pesticides and their usage [29]. In a recent survey of 208 farmers, about 47% of the participants stated that they felt unwell within 24 hours of pesticide use [30]. Workers in the sector are involved in practices where no preventive measures are undertaken and an acceptance that adverse health effects of agricultural work are expected [31]. In Trinidad and Tobago, agricultural injury and accident data is unavailable, magnifying the pesticide exposure predicament. There is also a lack of studies exploring the long-term effects of toxic pesticide use on farmers and consumers. The objective of this research is to examine the perception of the farmers, the representatives of the Agriculture Society of Trinidad and Tobago (ASTT), and the officials of the Ministry of Agriculture, Land and Fisheries (MALF) on the protection of farmers, particularly on the exposure to pesticides. The findings of this research will help contribute to policy-making by providing evidence from research on the state of the sector.

## Method

### Ethics statement

The study received approval from the Institutional Review Board of West Virginia University (Protocol #2401909110). No identifiers were collected during this study. A cover letter summarizing the study's objective and method was used to obtain the participant's written consent to participate in the research. By initialing at the end of the letter, the participant acknowledged that they had read and understood the study's purpose, scope, and voluntary nature.

### Study site

According to the World Bank, Trinidad and Tobago has a population of 1534937 with a population growth of 0.254 and a life expectancy of 74. Agricultural land accounts for 10.5% of the country's total land area, and of the total Gross Domestic Product (GDP) of Trinidad and Tobago, about 1.07% comes from agricultural activities [32]. In Trinidad and Tobago, the Ministry of Agriculture, Land and Fisheries (MALF) champions biodiversity conservation and sustainable food and non-food systems development. It is responsible for delivering integrated services aligned with a changing food and agriculture system [33]. Within the MALF exists an agency founded in 1839 named the Agriculture Society of Trinidad and Tobago (ASTT) that represents all farmers. It is a statutory body within the MALF representing all sectors, accounting for the 32774 farmers [34]. A total of 28442 farmers are located across eight counties in Trinidad and 4332 in the sister island of Tobago. Each county has multiple districts where an extension officer from the MALF is assigned to specific regional districts. An extension officer provides technical information on various areas, from the farmer registration process to agricultural technologies. Tobago is managed by the Division of Agriculture under the Tobago House of Assembly.

## Study design

The study is underscored using phenomenology. This approach explored the participants' experience, capturing their insights and attitudes towards pesticide use and its impact on their health. Interviews were conducted until data saturation. Thirty-two individuals from ASTT and MALF were invited to participate in the study voluntarily. The sample size was determined based on the literature recommendations, suggesting a minimum sample size between 12 and 30 participants as a standard for data [35]. A purposive sampling was used as participants were chosen due to their experience and knowledge in the sector specific to the study's objective.

The representatives of the MALF and ASTT were interviewed with approval from the Permanent Secretary. A request letter was sent to the Permanent Secretary on February 21st, 2024, and March 4th, 2024. All interviews were done face-to-face in person at the various offices of MALF from March 25th to April 12th, 2024. Participants from ASTT were interviewed from March 5th to March 20th in person and an online setting. The interviews lasted approximately 30–45 minutes each. In addition, one interview was conducted with a well-established local chemical manufacturer representative. This company was contacted on March 24th, and the interview was conducted on March 26th, face-to-face in person. All invitees agreed to participate in the study. All interviews were done privately with no interference.

## Data collection

An interview guide (Table 1) was created to assist in collecting data in the interview process.

## Interviewer

The field interviews were conducted by the author, a resident of the study site and a senior lecturer in occupational health and safety (OSH). The interviewer is a male and holds qualifications, namely an MSc in OSH and CertIOSH. He is a trained interviewer familiar with the local culture, language, and safety issues. Before the commencement of the interviews, the participants were informed of the researcher's professional background and the reasons for conducting the research. They were also notified of the importance of gathering information in the subject area due to the need to influence a healthier worker in a safe environment. A prepared letter was shared with the participants, giving an overview of the study and the protocol number stating the IRB approval. Sufficient time was also spent in the field, gaining more

Table 1. Interview guide.

| Leading Questions | Probing Questions |
|---|---|
| What is your overall assessment of the Agriculture Sector both as a viable sector and its importance to the economy of the country? | Describe the responsibility of your organisation as it pertains to the Agriculture Sector. |
| What are the Occupational Safety and Health challenges in the Agriculture Sector? | Are you aware of the ways to deal with these challenges? |
| What assistance, guidance and training do you provide to farmers? | How often are these training done? How do you determine what training interventions are needed? |
| What are the policies, systems or measures are in place to protect farmers against Exposure to pesticides, Pests, and Diseases | Are farmers aware of measures to deal with the issues of pesticide exposure, harmful chemicals and pests and diseases? Are farmers aware of proper ways to store and dispose of pesticides and other harmful chemicals? |
| Describe the specific safety regulations and guidelines in place for farmers? | Are there written standards in place to guide farmers? |
| How are safety standards monitored and enforced in agricultural operations? | What unique safety challenges are usually reported by farmers? How do you address these challenges? |
| What programs or initiatives exist to educate farmers about safety practices and procedures? | Are you aware when these education and training workshops are held? How is it held? Is it in person or online? Do you have access to an online platform for the online training? |

experience with the participants in their settings. The researcher developed an in-depth understanding, enabling the conveyance of details about the site and people, thus lending credibility. The interviews were conducted using a prepared interview protocol and recorded for further use in the data analysis process.

### Data analysis

All interviews were audio recorded with permission from the participants. The recordings were uploaded to a secure cloud server and transcribed using an automatic speech recognition system that enables transcription in multiple languages. The transcriptions were manually corrected for spelling and other errors. Then, identifiers such as names of people and specific places were removed from the text. The data were analyzed using an inductive thematic analysis [36].

The two researchers analyzed the data without predetermined categories, allowing themes to emerge directly from the participants' responses. This was done by reading and re-reading the interview transcripts, carefully examining the content for recurring words, ideas, and experiences. During this process, initial codes were created to capture patterns observed in the data, such as farmers' experiences with pesticide use, challenges accessing personal protective equipment, and concerns about health effects. As more transcripts were reviewed, these codes were refined and grouped into broader themes based on their similarities. For example, repeated mentions of discomfort and inconvenience in using PPE were categorized under the theme Barriers to PPE Adoption. After conducting inductive thematic analysis, deductive reasoning [37] was applied in the interpretation phase to contextualize and explain the findings within existing knowledge and literature. This process systematically compared the emerging themes with prior research, theoretical frameworks, and global best practices related to pesticide use, safety regulations, and agricultural occupational health.

Throughout, specific attention was given to the different perspectives of the stakeholders, such as the challenges farmers face with pesticide application, the regulatory and advisory roles of the Ministry of Agriculture, and the role of pesticide manufacturers in ensuring product safety and efficacy.

## Results

### Participants

A total of 32 interviews were conducted with participants representing both the ASTT and the MALF. A total of 17 participants were MALF Officers, 14 were farmers (ASTT), and 1 was a Technical Officer in a chemical company. Table 2 shows the participants' characteristics.

**Table 2. Participant characteristics (n = 32).**

|  | N(%) |
|---|---|
| **Sex** |  |
| Female | 19 (59) |
| Male | 13 (41) |
| **Age Category** |  |
| 18-35 | 6 (19) |
| 36-55 | 21 (66) |
| 56-65 | 2 (6) |
| 66 and over | 3 (9) |
| **Groups** |  |
| Ministry of Agriculture (MA) | 17 (53) |
| Farmer Group (FG) | 14 (44) |
| Chemical Company (CC) | 1 (3) |

## Themes

Responses from participants reveal several critical themes related to farming practices, chemical usage, and safety measures. Key themes include Barriers to Proper Use of Personal Protective Equipment (PPE), where participants highlight the challenges in adopting safety gear due to tradition, lack of knowledge, and urgency. The theme of Challenges in Safe Chemical Use and Disposal Practices emerges as participants discuss the misuse of chemicals and improper disposal methods that pose environmental risks. Another theme, Health Impacts of Unsafe Chemical Exposure, underscores the serious health consequences, such as cancer and skin diseases, linked to unsafe chemical handling. In Challenges in Farmer Training and Adoption of Safety Practices, participants point out issues with the timing, content, and engagement in training programs, particularly concerning health and safety. Lack of Policies and Monitoring for Farmer Safety and Chemical Use addresses the absence of regulatory frameworks and enforcement mechanisms to ensure safe practices. Finally, the theme of Absence of Quality Control and Unsafe Practices in Local Produce reveals concerns about the lack of monitoring for chemical residues in locally sold crops, with some participants favoring imported produce over local due to safety concerns. Together, these themes underscore the need for improved safety practices, training, policy development, and quality control in the agricultural sector.

## Theme 1: Barriers to Proper Use of Personal Protective Equipment (PPE) Among Farmers

All three stakeholder groups—farmers, the Ministry of Agriculture, and a representative from a pesticide manufacturing company—identified major barriers to the correct use of PPE. Common challenges included entrenched traditional practices, limited awareness of safety measures, inadequate education or training opportunities, and ingrained mindsets that underestimate the risks of pesticide exposure.

**Farmers' perspectives.** Farmers (FG) described scenarios in which protective measures were either haphazard or largely ignored.

> Participant 4 FG: *'There's nothing, you may see a farmer sometimes wear a jersey over his face and that's just because of the wind blow, you know, but there's not a lot of precautionary measures'*

Such accounts reflect a general lack of standardized or enforced PPE guidelines at the farm level. Similarly, Participant 6 (FG) highlighted both the diversity of suboptimal practices—such as avoiding masks, gloves, and long sleeves—and the potential for direct skin contact with chemicals when farmers walk in fields barefoot.

> Participant 6 FG: *'You still have farmers who will go there and not use the mask. You still have farmers who would not use a chemical mask or a dust mask. You still have farmers who would not use gloves. You still have farmers who would walk in the field bare feet. Most of them wouldn't wear a mask. They may wear a head cover. Some of them have a bandana or something like that. But they wear boots and not long pants. Some of them wear short sleeves. So they're still exposed'*

**Chemical company representative's perspective.** From the industry side, Participant 14 (CC) acknowledged that while pesticide companies often stress the importance of PPE, deep-seated traditions and generational habits hinder behavioral change:

> Participant 14 (CC): "We still stress the use of proper PPE… but based on tradition it's very difficult to change a person's mindset from doing things for years… spraying with the wind or against the wind without using respirators, masks, and even gloves."

**Ministry of agriculture's perspective.** Officials from the Ministry of Agriculture (MA) corroborated these observations, pointing to lack of information and resistance to new methods.

Participant 32 MA*: 'I think the number one, the number one problem I notice or feel is the incorrect use of pesticides and chemicals. The lack of problems farmers utilize, which is not using protective gear, proper PPE, more for better problems, most of them may not, they simply don't have the information, they lack the education, they lack the knowledge, they lack the advantages of using protective gear and some of them are just holding on to traditional based methods, which is, you know, simply put spraying chemicals without even respirators, without gloves, without boots and they do not see the importance of health and safety when they're on the field'*

### Theme 2: challenges in safe chemical use and disposal practices

Participants across all stakeholder groups recognized the misuse of chemicals, improper handling, and environmentally harmful disposal methods as widespread challenges. While farmers primarily spoke about on-the-ground realities, the Ministry of Agriculture officials emphasized the difficulty in monitoring and enforcing proper disposal, and no formal mechanism for chemical container collection was reported by any group.

**Farmers' perspectives.** Farmers observed that many of their peers show limited commitment to safe disposal and handling procedures. Participant 12 (FG) recounted seeing empty pesticide containers scattered around farmland and noted that while he knows bottles should be rinsed and punctured before disposal, many farmers do not follow these guidelines. Participant 14 (FG) similarly pointed out that storage conditions are often poor and that many farmers end up discarding used containers in rivers, contributing to both pollution and potential health hazards.

Participant 12 FG: '*So after using the chemical, they don't dispose of it properly. I, at least know you're supposed to rinse the bottle two to three times. And then you pierce the holes and this kind of thing. But from time to time, because as I told you, we have like about 20 farmers. When I go to visit and, help out some of the farmers, I often see these pesticides and weedicide bottles all over. It's just advertising*'

Participant 14 FG: '*Storage is one aspect, but how they store it is a different scenario because they have everything close to each other. And disposal, even disposal as well. For example, when you go down to most of these agriculture farms in large acreages, when they finish with that pesticide bottle, they throw it in the river, which leads to pollution and flooding. There's no organization or no company right now that I'm aware of that takes back empty pesticide plastic bottles to be recycled*'

**Ministry of agriculture's perspective.** Ministry representatives confirmed that these practices remain prevalent. Participant 16 (MA) explained the frequency of unsafe approaches:

Participant 16 MA: '*But in terms of safety aspects, you hardly see any type of change in terms of how they dress. Right. How they dispose and how they store the chemicals. They still store in open buckets or it's still open in the van tray. They still have bare feet. They still have no gloves*'

Participant 17 (MA) further underscored the routine sight of "empty bottles and chemical bottles" discarded on farms, while Participant 27 (MA) described confronting farmers directly about containers left in drains and rivers.

Participant 17 MA: '*When you go, you always see, empty bottles and chemical bottles, and I always encourage them, listen, try to dispose of these chemical bottles, don't leave them there, you know, because of the residual effect*'.

Participant 27 MA: *'Yes, you go out and you tell them, listen, these are your chemicals. Why are your bottles in the drain? Why is it in the river?'*

**Variation in awareness.** Despite these challenges, not all farmers neglect safe practices. Participant 2 (FG) demonstrated awareness of neutralizing chemical residues by washing containers in a charcoal pit, and Participant 6 (FG) mentioned deliberately reducing chemical use on certain crops, indicating that knowledge of safer methods does exist among some farmers.

Participant 2 FG: *'I'm aware that if I want to wash a can, you have a charcoal pit where you wash a can and through that access, that wastewater is washed away in the charcoal pit to neutralize the chemical'*

Participant 6 FG: *'The crops that they plant, you don't want to use many chemicals. Cocoa is not something that you use many chemical'*

**Theme 3: health impacts of unsafe chemical exposure**

Farmers from the focus groups shared firsthand observations and personal experiences illustrating the long-term health repercussions of unsafe pesticide practices. Ministry officials did not offer direct commentary on health impacts in this context, but their general concerns about PPE and disposal indirectly underscore the potential for severe health outcomes.

**Farmers' perspectives.** Several farmers recounted stories linking pesticide use to serious illnesses. Participant 1 (FG) cited rising rates of cancer and heart conditions among older farmers who have been exposed for decades.

Participant 1 FG: *'Yes, more or less the larger farmers who have been involved in agriculture since, let me say about 40 years ago, are now faced with cancer. You would see farmers from time to time having heart issues that are occurring right now'*

Participant 2 (FG) described a friend who repeatedly handled chemicals without protection and subsequently passed away at a young age.

Participant 2 FG: *'I know of a friend who would have passed late last year, a young guy. I always keep telling him the way I see him, the way he exposes himself to chemicals, no gloves, no this, no that. And he died just recently, so to say'*

The emotional weight of these experiences was also evident in Participant 5 (FG), who attributed a family member's death to unsafe pesticide handling.

Participant 5 FG: *'My mom died 12 years ago and I believe what brought it on, it didn't come on all of a sudden because they were misusing pesticides, they weren't using it safely'*

Moreover, Participant 12 (FG) highlighted farmers developing skin diseases yet not seeking medical help or recognizing the possible connection to chemical exposure.

Participant 12 FG: *'A lot of farmers have been getting skin disease and this kind of thing, and they're not taking it seriously or they're not even going to the doctor to verify if it is a result of the fertilizers being applied to their crops'*

These accounts reinforce the urgent need for enhanced safety measures and greater awareness of the potentially fatal health consequences linked to pesticide misuse.

                                                     

## Theme 4: challenges in farmer training and adoption of safety practices

Farmers and Ministry of Agriculture officials alike indicated that training programs—though occasionally available—are often poorly attended or insufficiently focused on health and safety. The agricultural extension services offer some form of instruction, but both groups reported that farmers' work schedules, skepticism, and a lack of dedicated health and safety content pose significant barriers.

**Farmers' perspectives.** Farmers acknowledged sporadic training opportunities but cited limitations. Participant 3 (FG) noted that many farmers remain unaware of scheduled sessions, while Participant 9 (FG) pointed out the impracticality of daytime classes since "that's prime work time for a farmer." Furthermore, Participant 8 (FG) remarked that the focus of most sessions is on crop production rather than on comprehensive health and safety measures.

Participant 3 FG: *'They might have one training and they will invite the farmers. Many of the farmers do not know'*

Participant 8 FG: *'From time to time they will offer training and then grow various crops. But to be honest, I have not seen anything catered specifically to target health and safety and the environment towards the farmers'*

Participant 9 FG: *'It is very challenging to be in a class from 9 o'clock to 11 o'clock and that's prime work time for a farmer, you know so having these classes doesn't make much of a difference'*

**Ministry of agriculture's perspective.** Officials confirmed these difficulties, with Participant 21 (MA) emphasizing that many farmers also hold regular jobs, making consistent attendance challenging.

Participant 21 MA: *most farmers now do have a regular job, so sometimes getting them out for training is a problem*

Participant 30 (MA) openly admitted that no formal safety and health training is provided.

Participant 30 MA: *'Not really, we do not do any training on safety and health'.*

Contrastingly, according to participant 25 (MA), they have sufficient safety training for farmers but they don't adhere to them.

Participant 25 MA: *'We do a lot of training in pesticide safety but some of them are hesitant because they think doing the correct thing is tedious for them'*

This gap is compounded by farmers' general reluctance to attend safety-focused sessions, seen as tedious or overly restrictive, as noted by Participant 12 (FG).

Participant 12 FG: *in terms of safety, full PPE, if you tell a farmer, listen, you need to come to our course so that we can teach you exactly what the full PPE that you should wear when applying these fertilizers the first thing they say is no, no, they don't want to hear that.*

## Theme 5: lack of policies and monitoring for farmer safety and chemical use

Farmers and Ministry of Agriculture officials jointly highlighted the absence of robust, formal policies or regulatory frameworks overseeing chemical usage, storage, and disposal. While extension officers can offer advice, there appears to be no enforcement mechanism or designated authority to monitor compliance.

**Farmers' perspectives.** Many farmers expressed a desire for clearer guidelines and accountability.

Participant 5 (FG), *"I would love to see that policy come about to help the safety of the farmers,"*

*Participant 15 FG: Based on research I've seen that farmers are suffering long-term from exposure to pesticides particularly because of lack of training lack of education poor farming practices from no monitoring and a lack of a policy*

Whereas Participant 8 (FG) stressed, "I don't think there is any safe use or safe practices for the agriculture sector in effect right now."

**Ministry of agriculture's perspective.** Ministry representatives conceded that current approaches rely largely on product labels.

Participant 16 MA: *'No, there is no policy or guideline in place for the safe use of chemicals, It's basically when you buy a product, there is a label and it's what the label tells you'*

While Participant 21 (MA) and 26 (MA) admitted they lack the authority to penalize noncompliance. This consensus across stakeholder groups underscores the need for a formalized policy to safeguard farmers' health and ensure responsible chemical management.

Participant 21 MA*: 'There's no policy in place with pesticide usage or anything like that'*

Participant 26 MA*: 'We don't have any responsibility and we cannot penalize the farmers for not following the guidelines'*

## Theme 6: absence of quality control and unsafe practices in local produce

Both farmers and Ministry officials voiced concerns about the lack of systematic monitoring of chemical residues in locally grown crops. While export-oriented produce undergoes more stringent checks, domestic markets remain largely unregulated, creating potential risks for consumer health, including the farmers themselves.

**Farmers' perspectives.** Some farmers explicitly avoid consuming locally grown produce. Participant 7 (FG) explained cautioning family members to purchase imported cabbage, fearing pesticide residues in local alternatives.

Participant 7 FG: *I just tell my parents or my friends and siblings, when they buy leafy vegetables, especially cabbage, don't buy from Trinidad, buy the imported one"*

Participant 12 (FG) emphasized the need for random testing of market produce to deter unsafe use of chemicals, suggesting "some fear" might motivate farmers to observe pre-harvest intervals more diligently.

*Participant 12 FG: I think that more has to be done to ensure that when farmers use whatever chemical they do following the pre-harvest interval period. I don't know how they will do it, but they need to have something testing crops, at least randomly testing the crops in the market. And so at least instil some fear in the farmer to know and listen. If your crop tests positive for some chemical that is not supposed to be utilized, there must be some form of penalty.*

**Ministry of agriculture's perspective.** Ministry officials confirmed that misuse persists, with farmers sometimes applying non-approved or high-toxicity chemicals close to harvest.

*Participant 16 MA: 'Some farmers want to spray the black disinfectant. Farmers do their own thing. You're telling them not to use products that are not designed for crops, especially on stuff that you use fresh. But they are persistent'*

Participant 27 (MA) recounted a farmer knowingly using large quantities of chemicals on cabbage, acknowledging it was dangerous but proceeding anyway.

> Participant 27 MA: *'I would have gone and the farmer had his cabbage with some other worm bites. So he's fully well aware that the amount of chemicals he's using is dangerous to himself and even others. So he chose to eat it and sell it to the grocery stores as well'*

These accounts highlight a critical regulatory gap that compromises food safety and underscores the broader need for stronger enforcement and monitoring mechanisms.

**Chemical company's perspective.** The chemical company representative confirmed that many farmers do not follow the required waiting periods before harvesting.

Participant 14 (CC) noted, "A lot of the farmers actually would not even observe the pre-harvest interval times of pesticides anyhow so they will use products that have long harvest intervals and harvest the next two or three days which generally for the consumers is very toxic and very harmful to them."

## Summary of results

Across all themes, a consistent finding is the interplay between limited safety awareness, entrenched cultural practices, and the absence of rigorous policy enforcement. Farmers, Ministry of Agriculture officials, and industry representatives all acknowledge that while isolated instances of good practice exist, these are overshadowed by ongoing barriers to effective training, consistent use of PPE, proper chemical handling, and thorough regulatory oversight. Consequently, farmers remain at significant risk of adverse health outcomes, and consumers may face potential hazards from unmonitored chemical residues in locally produced food.

## Discussion

### Main findings and contribution

This study makes a significant contribution as the first to explore the impact of pesticide use on farmer health and safety in Trinidad and Tobago, gathering perspectives from farmers and the Ministry of Agriculture. The themes emerging from the findings reveal an interconnected system of challenges that contribute to unsafe pesticide use, inadequate safety measures, and long-term health risks for farmers and consumers. At the core of these challenges lies the absence of a structured policy framework and enforcement mechanisms, which is the root cause of the gaps identified across training, cultural practices, chemical handling, and health outcomes. The lack of formal policies regulating pesticide use, protective measures, and disposal practices results in limited oversight, leaving farmers responsible for their safety decisions without institutional guidance or regulatory accountability.

Ministry of Agriculture officials acknowledge this policy vacuum, emphasizing that there is no regulatory mechanism to monitor compliance, penalize unsafe practices, or ensure adherence to pesticide safety guidelines beyond what is printed on product labels. Farmers continue to operate without clear mandates in an environment where pesticide safety is discretionary rather than obligatory. This absence of enforcement directly affects training and education, as farmers are not required to undergo pesticide safety training. Although the Ministry of Agriculture and other agricultural organizations provide some training opportunities, they are not widely attended due to logistical constraints, such as scheduling conflicts with peak farming hours and a general reluctance among farmers to engage in safety-focused sessions.

Farmers often perceive such training as tedious or unnecessary, particularly when it does not meet their immediate economic needs. Even when training is provided, it is usually limited to crop production techniques rather than comprehensive occupational safety and health education. The lack of a regulatory push for structured training programs results in

knowledge gaps and inconsistent safety practices, making farmers more susceptible to adopting risky pesticide-handling behaviors.

Compounding the issue of inadequate education is the persistence of traditional and cultural farming practices, which significantly influence the adoption of safety measures. Many farmers have inherited generational farming techniques prioritizing efficiency and convenience over safety. As a result, the use of PPE is often inconsistent, with some farmers relying on minimal protection, such as using a bandana instead of a respirator or walking barefoot in pesticide-treated fields. Even when PPE is available, farmers frequently report discomfort, inconvenience, or a lack of habit in using protective gear. The resistance to change is further reinforced by a belief that existing practices are sufficient despite increasing evidence of adverse health effects. These entrenched cultural norms are barriers to behavior change, making it difficult for training programs and safety recommendations to take hold. The inadequate use of PPE and the persistence of traditional pesticide-handling methods contribute to unsafe chemical application and disposal practices, leading to direct environmental and health hazards.

Farmers often misapply pesticides, fail to observe pre-harvest intervals, and dispose of empty chemical containers in rivers, fields, or drainage systems. The lack of a structured disposal system exacerbates this issue, as there are no established collection or recycling programs for empty pesticide bottles. Ministry officials express concern over the widespread disposal of hazardous chemicals in the environment, noting that no designated agency is responsible for managing or monitoring the safe handling of pesticide waste. The improper use and disposal of pesticides heighten farmers' exposure to toxic substances and pose risks to surrounding communities through the contamination of water sources and soil degradation.

Many farmers in the study reported long-term health deterioration among themselves or their peers, attributing these conditions to prolonged pesticide exposure without adequate protection. The health risks are further aggravated by the absence of medical intervention and awareness, as farmers often do not seek medical attention or do not associate their symptoms with pesticide exposure. This lack of recognition of pesticide-related illnesses contributes to continued unsafe practices, creating a cycle of exposure and declining health that remains unaddressed due to weak regulatory oversight.

Beyond individual health risks, these unsafe farming practices also affect food safety and consumer health, mainly due to the absence of quality control and monitoring of pesticide residues in local produce. While export-oriented agricultural products undergo stringent checks, produce intended for local consumption remains largely unregulated. Farmers and ministry officials acknowledge that many farmers do not observe pre-harvest intervals, meaning that pesticide-treated crops are harvested and sold with potentially toxic chemical residues still existing. The lack of enforcement in this area raises significant concerns about public health, as consumers unknowingly purchase and consume food that may contain harmful levels of pesticide contamination. This, in turn, has led to a loss of confidence in locally grown produce, with some farmers themselves opting to buy imported vegetables rather than consume their pesticide-treated crops.

## Comparison with the literature

**PPE use.** Farmers' negligence towards PPE, as noted in our study, is consistent with the literature. Widianto et al. identify several internal factors that affect farmers' ability to recognize hazardous materials, including their age, farming experience, and education level [38]. Similarly, Imoro et al. report that many farmers do not utilize safety gloves or masks during pesticide application, leading to significant health risks from dermal and inhalation exposure [39]. Moreover, the literature indicates that compliance remains low even when farmers know the necessity of PPE. For instance, Sai et al. found that while many farmers recognized the need for protective gear, actual usage was inconsistent [40]. This finding resonates with our observation that despite the availability of PPE, its use is often compromised by convenience and habitual practices.

Furthermore, Odisho and Mohammad highlight that improper PPE usage and unsafe practices like eating or drinking on the farm significantly increase the risk of pesticide exposure [41]. This suggests that mere availability of PPE is

insufficient; there must be a concerted effort to change farmers' mindsets and behaviors towards its use. The literature also emphasizes the role of education in improving PPE compliance. For example, Rosanti et al. advocate for comprehensive education on pesticide safety management, which includes proper PPE usage [42]. This is echoed by Damalas and Koutroubas, who assert that training on pesticide use is associated with elevated safety behaviors among farmers [14]. Our study supports this notion, indicating that more targeted educational interventions are necessary to foster a safety culture among farmers. However, some research suggests a positive correlation between education and PPE usage, while others report no significant relationship [43,44].

**Pesticide use and safe handling.** Participants in our study indicated a tendency to misuse pesticides, particularly by neglecting pre-harvest intervals, and to dispose of chemical containers unsafely. This aligns with existing literature highlighting similar behavior patterns among farmers in various regions. Research by Sai et al. emphasizes that improper pesticide disposal contaminates soil and water sources, adversely affecting non-target organisms and the broader ecosystem [40]. Similarly, Mergia et al. corroborate these findings, noting that the disposal of leftover pesticides and empty containers is a critical concern, particularly in LMICs [45]. This is echoed by Hashemi et al.., who point out that inadequate training and awareness among farmers contribute to unsafe pesticide use and disposal practices, leading to significant health risks [46]. Hashemi et al. also found that farmers with higher education levels exhibited better knowledge and safer behaviors regarding pesticide use [46]. This suggests that targeted educational programs could mitigate the risks associated with improper pesticide handling, as highlighted by multiple other studies [47,48].

The disposal methods reported by participants in our study—such as burning or burying empty containers—are consistent with findings from other regions. For example, Jallow et al. noted that many farmers in Kuwait also adopted hazardous disposal methods, leading to environmental contamination [21]. Such practices threaten environmental integrity and pose direct health risks to farmers and consumers alike, as improper disposal can lead to pesticide residues entering the food chain.

## Health risks

Our findings regarding the health impacts of unsafe chemical exposure among farmers reveal a concerning trend of severe health conditions, including cancer, heart disease, and skin disorders, attributed to inadequate safety measures and unsafe practices. This aligns closely with the existing literature, consistently highlighting the adverse health effects of pesticide exposure in agricultural settings [23]. Karunamoorthi et al. report that many farmers are exposed to harmful pesticides, which are often banned in industrialized countries, further compounding health risks [49].

In addition, the literature suggests that the cumulative effects of long-term pesticide exposure can lead to chronic health conditions. For example, Fibriansari et al. note that farmers with prolonged pesticide exposure often do not use PPE, resulting in a decline in their immune function and increased susceptibility to various diseases [50]. This aligns with our findings, where participants reported serious health conditions that may be linked to long-term pesticide exposure without adequate protective measures.

Our study also highlights the safety knowledge and training gap among farmers, highlighting the importance of targeted educational interventions to mitigate health risks. However, some studies suggest that awareness of pesticide hazards does not always translate into safe practices. For instance, Pengpan found that even when farmers are aware of the risks associated with pesticide exposure, they often fail to implement protective measures effectively [51]. This complexity indicates that simply increasing awareness may not be sufficient; practical solutions and behavioral changes are necessary to improve health outcomes among farmers.

## Safety training and policy

The findings regarding the ineffectiveness of training and education programs for farmers, particularly concerning the safe use of chemicals, highlight significant barriers to participation and engagement. Despite the efforts of extension officers and organizations to provide training, issues such as inconvenient schedules, perceptions of safety measures as tedious,

and reluctance to attend safety-focused sessions persist. This situation calls for a reassessment of training approaches to better align with farmers' needs. The literature supports these findings, indicating that training programs often fail to achieve their intended outcomes due to similar barriers. For instance, Rahaman et al. emphasize the importance of motivating farmers and extension staff to enhance pesticide safety practices [52]. They argue that training programs will continue to fall short without addressing the underlying barriers to participation, such as scheduling and perceived relevance [52]. Similarly, Tandi et al. note that many farmers lack knowledge about appropriate pesticide management techniques, which can be exacerbated by poorly designed training programs that do not consider farmers' practical realities [53]. This suggests that considering the farmers' context and preferences, a more tailored approach to education is essential for improving engagement and compliance with safety measures.

As noted in our study, the absence of safety policies and the need for robust monitoring systems for the safe use of chemicals and farmer health resonates with the literature. Nicolopoulou-Stamati et al. argue for an agricultural concept that emphasizes reducing chemical pesticide use and implementing effective monitoring systems to safeguard human health and the environment [54]. This aligns with the findings that farmers are left to manage risks independently without clear guidelines and enforcement mechanisms, often prioritizing productivity over safety. Furthermore, Wu et al. discuss the need for improved environmental policy regulation and enforcement to address the overuse of agricultural chemicals [55]. They suggest that without adequate policies and monitoring, the risks associated with chemical use will continue to escalate [55]. This is echoed by Jepson et al., who highlight the widespread and severe risks associated with pesticide use, emphasizing the need for an enabling environment that supports sustainable agricultural practices [56].

## Food safety and use of toxic chemicals

Our findings regarding the lack of quality control in locally sold produce and the resulting food safety concerns highlight a significant issue in consumer confidence. Participants preferred imported foods, believing them to be safer than locally grown crops, underscoring the need for enhanced quality assurance measures for local crops. This situation is consistent with existing literature that addresses consumer perceptions of food safety and the implications for local agricultural practices. Research by Scarpato et al. emphasizes that consumers are increasingly concerned about food safety, which significantly influences their purchasing decisions [57]. They argue that policy strategies are essential to reassure consumers about the safety of food products [57]. This aligns with our findings, where the lack of routine testing for chemical residues in locally grown crops has diminished consumer trust. The perception that imported goods are safer reflects a broader trend where consumers prioritize safety over local sourcing, as Obayelu et al. noted that health risks associated with toxic pesticide use in vegetable production contribute to consumer hesitance regarding local produce [58].

Moreover, the literature indicates that the absence of quality control measures can lead to significant public health risks. For instance, Xi et al. discuss the increasing concern over food safety incidents and the need for regulatory authorities to implement stringent quality and safety supervision in agricultural production [59]. This finding resonates with our study's emphasis on the necessity for robust quality assurance measures to ensure that both local and export crops meet safety standards.

## Recommendations

The safety concerns identified in our study call for a multi-level intervention, targeting policy reform, education, infrastructure, and monitoring mechanisms. However, it is important to acknowledge that many of these interventions require financial and institutional resources that may not always be readily available, particularly in LMICs where agricultural funding and regulatory capacity are often constrained. Therefore, careful prioritization, phased implementation, and innovative partnerships will be essential to achieving meaningful progress under resource-limited conditions.

### Strengthening policy and regulatory frameworks

A critical first step in improving farmer safety is establishing enforceable policies that regulate pesticide use, PPE adoption, and safe disposal practices. National or regional regulations should mandate using PPE when handling pesticides, with specific guidelines for respirators, gloves, and boots. Furthermore, enforcing pre-harvest interval (PHI) compliance through randomized pesticide residue testing in local markets can help ensure that farmers follow recommended waiting periods before harvesting pesticide-treated crops [60,61]. To address the issue of pesticide container disposal, formal chemical disposal guidelines should require manufacturers and distributors to implement take-back programs [62,63], providing designated collection points for used containers. In addition to regulatory measures, a farm safety certification program should be introduced, linking compliance with safety regulations to incentives such as government subsidies, agricultural loans, or tax reductions [64]. This would encourage farmers to adhere to safety protocols while providing economic benefits for their compliance. Strengthening monitoring and inspection units within agricultural extension services is also essential to effectively implement these policies.

### Expanding and improving training programs

Training is crucial in enhancing farmers' awareness and adoption of safe pesticide practices. Mandatory pesticide safety training should be implemented to increase participation, requiring farmers to complete annual sessions before purchasing or using high-risk pesticides. These training sessions should be conducted on-site rather than in classrooms to ensure hands-on learning in real farming environments. Moreover, on-site demonstration programs can give farmers practical exposure to proper pesticide handling, PPE use, and chemical disposal techniques. Farmers who complete training programs should receive subsidized or free [65] PPE kits to incentivize participation further, reducing the financial barrier to adopting protective measures.

Establishing farmer-to-farmer training networks [66] can also enhance knowledge dissemination, wherein lead farmers are trained to educate their peers, fostering community-driven safety awareness. Mobile training units can also be deployed to remote farming areas, ensuring that safety education reaches all farmers, regardless of location.

### Addressing cultural and traditional barriers

Many farmers have inherited generational farming methods prioritizing efficiency over safety, leading to the continued reliance on unsafe pesticide application and minimal protective gear. Community influencers and respected local farmers should be engaged as safety advocates to shift ingrained behaviors, reinforcing the importance of protective practices through peer-led initiatives.

Recognizing that farmers may resist abrupt changes, training programs should integrate traditional and modern hybrid solutions, acknowledging existing farming knowledge while introducing practical, science-based safety enhancements. A long-term approach to cultural change should include youth engagement in agricultural safety education, ensuring that the next generation of farmers is better equipped with safety knowledge.

### User-centric PPE design

Standard PPE may be uncomfortable in hot, humid climates, leading to non-compliance. Partnering with manufacturers to develop breathable, lightweight PPE suited for tropical conditions can enhance adoption rates. Retail distribution strategies should also be improved by requiring pesticide sellers to bundle PPE with pesticide purchases, ensuring that protective equipment is readily available at the point of sale. Finally, proper PPE use, maintenance, and storage training should be integrated into pesticide safety programs to ensure that equipment is used effectively and remains in good condition.

### Limitations

One limitation of our study is that it relied on self-reported data, which may be subject to recall bias or selective reporting. Participants may not have fully disclosed unsafe practices or health concerns due to social desirability bias, leading

to potential underreporting of issues related to chemical exposure. Additionally, our study focused on a specific group of farmers and Ministry of Agriculture representatives in Trinidad and Tobago, which may limit the generalizability of the findings to other regions or contexts with different agricultural practices and regulatory environments. The absence of triangulation with quantitative data or direct observation of farming practices further constrains the study's ability to verify participants' accounts.

## Conclusions

In conclusion, the findings and literature collectively emphasize the urgent need for comprehensive strategies that integrate health and safety training, effective policy implementation, and consumer education to address the challenges faced in agricultural practices. By prioritizing these areas, stakeholders can work towards creating a safer and more sustainable agricultural landscape that benefits farmers and consumers alike.

### Protection of human and animal subjects

No human subjects were involved in the project.

### Author contributions

**Conceptualization:** Avishek Choudhury.

**Data curation:** Barry Parasram.

**Formal analysis:** Barry Parasram, Avishek Choudhury.

**Investigation:** Barry Parasram, Avishek Choudhury.

**Methodology:** Barry Parasram, Avishek Choudhury.

**Resources:** Avishek Choudhury.

**Supervision:** Avishek Choudhury.

**Validation:** Avishek Choudhury.

**Writing – original draft:** Barry Parasram, Avishek Choudhury.

**Writing – review & editing:** Avishek Choudhury.

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
