## [Decision Letter · Decision Letter 0]

PONE-D-24-46271Occupational Health Risk of Farmers: A Qualitative Study With the Agriculture Society of Trinidad and Tobago and the Ministry of Agriculture, Land and Fisheries.PLOS ONE

Dear Dr. Choudhury,

Thank you for submitting your manuscript to PLOS ONE. After careful consideration, we feel that it has merit but does not fully meet PLOS ONE’s publication criteria as it currently stands. Therefore, we invite you to submit a revised version of the manuscript that addresses the points raised during the review process.

**ACADEMIC EDITOR: Experts in the field have reviewed your manuscript and you are expected to address their comments as early as possible. Thank you.>==============================**

**A rebuttal letter that responds to each point raised by the academic editor and reviewer(s). You should upload this letter as a separate file labeled 'Response to Reviewers'.****A marked-up copy of your manuscript that highlights changes made to the original version. You should upload this as a separate file labeled 'Revised Manuscript with Track Changes'.****An unmarked version of your revised paper without tracked changes. You should upload this as a separate file labeled 'Manuscript'.**

****

We look forward to receiving your revised manuscript.

**Kind regards,**

**Olutosin Ademola Otekunrin**

Academic Editor

PLOS ONE

3. Please note that your Data Availability Statement is currently missing [the repository name and/or the DOI/accession number of each dataset OR a direct link to access each database]. If your manuscript is accepted for publication, you will be asked to provide these details on a very short timeline. We therefore suggest that you provide this information now, though we will not hold up the peer review process if you are unable. 

**Additional Editor Comments (if provided):**

Reviewers' comments:

Reviewer's Responses to Questions

**Comments to the Author**

1. Is the manuscript technically sound, and do the data support the conclusions?

**Reviewer #1: Yes**

**Reviewer #2: Yes**

**2. Has the statistical analysis been performed appropriately and rigorously? **

**Reviewer #1: Yes**

**Reviewer #2: Yes**

3. Have the authors made all data underlying the findings in their manuscript fully available?

**Reviewer #1: Yes**

**Reviewer #2: No**

4. Is the manuscript presented in an intelligible fashion and written in standard English?

**Reviewer #1: Yes**

**Reviewer #2: Yes**

5. Review Comments to the Author

**Reviewer #1: The manuscript addresses a critical topic, occupational health risks for farmers. While the findings are relevant and contribute to the field, there are areas needing improvement for clarity and impact;**

1. The introduction should better connect the study to global contexts by referencing similar research in other regions.

2. The methodology section requires justification of the sample size and details on confidentiality measures.

3. The results section would benefit from clearer differentiation between stakeholder perspectives and more contextual explanations of participant quotes.

4. Recommendations should include specific, actionable interventions to enhance farmer safety.

Consider expanding the discussion on discrepancies with existing literature and addressing potential biases due to self-reported data.

**Reviewer #2: Strength of the manuscript**

The manuscript addresses a crucial public health issue by examining occupational health risks among farmers due to pesticide exposure. Its focus on Trinidad and Tobago provides valuable regional insight. By incorporating interviews with participants from various stakeholder groups involved in pesticide use, the study offers a comprehensive and in-depth analysis.

Things to improve

1.Line 107 and 108 – “Its 10.5% of the land area is designated as agricultural land with a contribution of 1.07% towards GDP.” This sentence need clarity.

2.Line 123 – which problem was identified and which area of research?

3.Did the participants provide signed consent?

4.How was inductive and deductive phase of coding done?

5.How can the result in this study serve as a basis for further quantitative research?

6. PLOS authors have the option to publish the peer review history of their article (what does this mean? ). If published, this will include your full peer review and any attached files.

**Do you want your identity to be public for this peer review?** For information about this choice, including consent withdrawal, please see our Privacy Policy . 

**Reviewer #1: **Yes: ** Bukade Adesina**

**Reviewer #2: No**

****

---

## [Author Response · Author response to Decision Letter 1]

20 Feb 2025

Please see the attached response document for better formatting.

Reviewer #1

The manuscript addresses a critical topic, occupational health risks for farmers. While the findings are relevant and contribute to the field, there are areas needing improvement for clarity and impact.

Response: Thank you for all the constructive feedback. We have made all the changes suggested.

1. The introduction should better connect the study to global contexts by referencing similar research in other regions.

Response: We have extensively revised the introduction and added literature review as per your suggestions.

“For instance, reductions in cholinesterase activity—an indicator of neurotoxic effects—have been observed in pesticide applicators (Nambunmee et al., 2021). Chronic exposure has been associated with dermatitis, respiratory issues, and neurological deficits, as reported among Thai rice farmers (Sapbamrer & Nata, 2013). The repercussions of pesticide overuse extend beyond individual farmers to entire communities and ecosystems. Contamination of water sources and threats to food safety have been documented in Bangladesh (Kobir et al., 2020) and in other regions with high pesticide reliance (Bhattacharjee et al., 2013). Such community-wide health risks accentuate the need for policies that safeguard environmental integrity alongside farmer well-being.

Safety training is consistently cited as critical interventions for reducing pesticide-related risks. Farmers who receive targeted training are more likely to engage in safer pesticide handling and application (Damalas & Koutroubas, 2017). Ren and Jiang reported that agricultural cooperatives' training in Shandong Province, China, led to a decrease in pesticide overuse among farmers, emphasizing the role of structured training in promoting safe practices (Ren & Jiang, 2022). Integrated Pest Management (IPM) training has emerged as an effective strategy for reducing pesticide exposure. Clausen et al. demonstrated that IPM training in Uganda resulted in lower organophosphate exposure among farmers, as measured by acetylcholinesterase (AChE) blood levels (Clausen et al., 2017). This suggests that IPM not only promotes sustainable agricultural practices but also enhances the health and safety of farmers. Furthermore, Son et al. found that after two years of IPM training in Burkina Faso, farmers used less pesticide and experienced reduced exposure, indicating the long-term benefits of such educational interventions (Son et al., 2018).

However, access to effective training remains limited in many LMICs. Nwadike et al. highlighted that many farmers in Northern Nigeria lack sufficient occupational safety knowledge, which contributes to unsafe pesticide application practices Nwadike et al. (2021). LMICs often lack safety training and or policies and their implementation due to limited resources 1010. This finding is echoed by Çevik et al., who reported that farmers' knowledge, attitudes, and practices regarding pesticide safety are often insufficient, indicating a pressing need for targeted training interventions (Çevik et al., 2023). Despite some awareness of the risks associated with pesticide use, many farmers fail to implement safe practices, suggesting that knowledge alone is not enough to change behavior (Jallow et al., 2017). In Tanzania, farmers were aware of how pesticides could be absorbed, yet they did not consistently translate this knowledge into safer practices (Lekei et al., 2014). Proper use of PPE is also essential to minimize pesticide exposure. Yet research indicates that farmers often do not use PPE correctly, do not have reliable access to it, or discontinue its use (Kapeleka et al., 2019; Pandiyan et al., 2023; Okoffo et al., 2016). Even when farmers are conscious of the risks, their PPE usage may not effectively reduce exposure if the equipment is of low quality or if training on its proper use is inadequate (Kapeleka et al., 2019).

Governmental policies and regulations are essential for ensuring that farmers have access to safe pesticides and training resources. Phung et al. emphasized the need for improved pesticide regulations in Vietnam to enhance farmworker safety, highlighting that many farmers are unfamiliar with safety practices and regulations (Phung et al., 2012). Similarly, Fuhrimann et al. pointed out that the lack of effective legislative frameworks in LMICs contributes to the high prevalence of pesticide-related health issues (Fuhrimann et al., 2019). Therefore, strengthening regulatory frameworks and providing support for compliance is vital for improving pesticide safety among farmers. Moreover, the socio-economic context of farmers significantly influences their ability to engage in safe pesticide practices. Many farmers operate under economic constraints that prioritize immediate productivity over long-term health and safety. As noted by Khan et al., the lack of information and awareness about pesticide hazards often leads to overuse and misuse of these chemicals, exacerbating health risks (Khan et al., 2019).”

2. The methodology section requires justification of the sample size and details on confidentiality measures.

Response: We have clarified this in the text. The literature suggests that the minimum sample size for qualitative interview studies typically ranges from 12 to 30 participants, with many studies recommending around 25-30 as a standard for achieving data saturation. In a more recent study, Kohut et al. indicated that a minimum goal of 30 interviews was deemed feasible for their exploratory analysis, reinforcing the idea that sample sizes between 20 and 30 are often recommended for qualitative stud

ies (Kohut et al., 2023). This is echoed by Wutich et al., who conducted an integrative review and found that sample sizes for qualitative data analysis typically range from 20 to 40 interviews, depending on the study's objectives and design (Wutich et al., 2024).

Additionally, The concept of "information power," as proposed by Malterud et al., suggests that the adequacy of sample size is not solely dependent on the number of participants but also on the richness of the data collected. They argue that a smaller sample size may be sufficient if the participants provide detailed and relevant information (Malterud et al., 2016). This perspective encourages researchers to focus on the quality of data rather than merely the quantity of interviews.

References:

Kohut, O., Wang, Z., Sanchez, R., Rausch, J., Nieto, A., & Minguez, M. (2023). Assessing the impact of a 6-year health sciences enrichment program for underrepresented minority youth on healthcare workforce diversity, career path, and public health. Frontiers in Public Health, 11. https://doi.org/10.3389/fpubh.2023.1244593

Wutich, A., Beresford, M., & Bernard, H. (2024). Sample sizes for 10 types of qualitative data analysis: an integrative review, empirical guidance, and next steps. International Journal of Qualitative Methods, 23. https://doi.org/10.1177/16094069241296206

Regarding confidentiality, the audio recordings were transcribed and manually deidentified. Recordings were then deleted. This is a standard protocol.

See Data Analysis Section

“All interviews were audio recorded with permission from the participants. The recordings were uploaded to a secure cloud server and each was transcribed using the software ‘Whisper’18. It is an automatic speech recognition system that enables transcription in multiple languages. The transcriptions were also manually corrected for spelling and other errors. Then, identifiers such as names of persons and places were removed from the text.”

3. The results section would benefit from clearer differentiation between stakeholder perspectives and more contextual explanations of participant quotes.

Response: We have restructured the entire result section. We have now separated and compared all stakeholder groups. Please see the result section.

4. Recommendations should include specific, actionable interventions to enhance farmer safety.

Response: We agree. We have added an elaborate and practical recommendation section. We have also acknowledged that these recommendations might need stakeholder to prioritize before implementing based on availability of resources and affordability. Please the Recommendation section.

5. Consider expanding the discussion on discrepancies with existing literature and addressing potential biases due to self-reported data.

Response: Thank you for this comment. We have added new section to the discussion. Comparing with literature was already there, additionally we have expanded on the findings and implications. Please see the Main Contribution section.

We also had the limitations about self-reported biases in the discussion section. Please see the limitation section.

Reviewer #2

The manuscript addresses a crucial public health issue by examining occupational health risks among farmers due to pesticide exposure. Its focus on Trinidad and Tobago provides valuable regional insight. By incorporating interviews with participants from various stakeholder groups involved in pesticide use, the study offers a comprehensive and in-depth analysis.

Response: Thank you for all the constructive feedback. We have addressed all your comments and made substantial changes throughout the manuscript.

1.Line 107 and 108 – “Its 10.5% of the land area is designated as agricultural land with a contribution of 1.07% towards GDP.” This sentence need clarity.

Response: We have rephrased this for clarity.

“Agricultural land accounts for 10.5% of the country's total land area and of the total Gross Domestic Product (GDP) of Trinidad and Tobago, about 1.07% comes from agricultural activities”

2.Line 123 – which problem was identified and which area of research?

Response: We deleted this sentence to avoid any confusion.

3.Did the participants provide signed consent?

Response: Yes. They gave their consent by writing their initials on the cover letter. We have now clarified this in the manuscript. The cover letter sample was also shared with PLOS journal.

“The study received approval from the Institutional Review Board of West Virginia University (Protocol #2401909110). No identifiers were collected during this study. A cover letter summarizing the objective and method of the study was used for obtaining the participant’s written consent to take part in the research. By initialing at the end of the letter, the participant acknowledged that they had read and understood the purpose, scope, and voluntary nature of the study.”

4.How was inductive and deductive phase of coding done?

Response: We actually did inductive thematic analyses with deductive reasoning. We have rectified this in the manuscript. For inductive thematic analysis, we used the standard practice. The inductive phase of coding was conducted without predefined categories or themes, allowing patterns to emerge naturally from the interview transcripts. In this phase, two authors independently reviewed the transcripts in detail, engaging in a line-by-line analysis to capture the meaning and context of participants' responses. Each interview was carefully examined to identify recurring words, phrases, and ideas, which were then grouped into preliminary codes that reflected key issues raised by participants. For instance, when farmers discussed their reluctance to use personal protective equipment (PPE), common phrases such as "too hot to wear," "not necessary," or "never used it before" began to appear frequently. These recurring ideas were coded under an emerging theme related to barriers to PPE adoption. Similarly, when multiple participants from the Ministry of Agriculture described the lack of enforcement in pesticide regulation, a new theme related to weak regulatory oversight was formed.

After the inductive coding process established the initial set of themes, the deductive reasoning was used to structure and refine these findings based on existing theoretical frameworks and prior research on pesticide safety and occupational health.

5.How can the result in this study serve as a basis for further quantitative research?

Response: This is our ongoing study. We are administering survey with approximately 1000 farmers and focusing on quantifying the risks identified during the interviews. Additionally, we are also extending to physical harms like MSDs, Mental stress, fatigue, and non-fatal injuries.

We expect to complete this data collection by this year end.

---

## [Editor Report · Decision Letter 1]

Occupational Health Risk of Farmers: A Qualitative Study With the Agriculture Society of Trinidad and Tobago and the Ministry of Agriculture, Land and Fisheries.

PONE-D-24-46271R1

Dear Dr. Choudhury,

We’re pleased to inform you that your manuscript has been judged scientifically suitable for publication and will be formally accepted for publication once it meets all outstanding technical requirements.

Kind regards,

Olutosin Ademola Otekunrin

Academic Editor

PLOS ONE
---

## [Editor Report · Acceptance letter]

PONE-D-24-46271R1

PLOS ONE

Dear Dr. Choudhury,

I'm pleased to inform you that your manuscript has been deemed suitable for publication in PLOS ONE. Congratulations! Your manuscript is now being handed over to our production team.

Kind regards,

on behalf of

Dr. Olutosin Ademola Otekunrin

Academic Editor

PLOS ONE